



# Methanethiol, dimethyl sulfide and acetone over biologically
# productive waters in the SW Pacific Ocean
Sarah J. Lawson[1], Cliff S. Law[2,3], Mike J. Harvey[2], Tom G. Bell[4], Carolyn F. Walker[2], Warren
J. de Bruyn[5] and Eric S. Saltzman[6]
[1] Commonwealth Scientific and Industrial Research Organisation, Oceans and Atmosphere, Aspendale, Australia
[2] National Institute of Water and Atmospheric Research, Wellington, New Zealand
[3] Dept. Chemistry, University of Otago, Dunedin, New Zealand
[4] Plymouth Marine Laboratory, Plymouth, UK
[5] Schmidt College of Science and Technology, Chapman University, Orange, California, CA, USA
[6] Earth System Science, University of California, Irvine, California, USA
*Correspondence to*: Sarah J. Lawson (sarah.lawson@csiro.au)
**Abstract**
Atmospheric methanethiol ($MeSH_a$), dimethyl sulfide ($DMS_a$) and acetone ($acetone_a$) were measured over
biologically productive frontal waters in the remote South West Pacific Ocean in summertime 2012 during the
Surface Ocean Aerosol Production (SOAP) voyage. $MeSH_a$ mixing ratios varied from below detection limit (< 10
ppt) up to 65 ppt and were 3 - 36% of parallel $DMS_a$ mixing ratios. $MeSH_a$ and $DMS_a$ were correlated over the
voyage ($R^2$ = 0.3, slope = 0.07) with a stronger correlation over a coccolithophore-dominated phytoplankton bloom
($R^2$ = 0.5, slope 0.13). The diurnal cycle for $MeSH_a$ shows similar behaviour to $DMS_a$ with mixing ratios varying
by a factor of ~2 according to time of day with the minimum levels of both $MeSH_a$ and $DMS_a$ occurring at around
16:00 hrs. A positive flux of MeSH was calculated for 3 different nights and ranged from 3.5 - 5.8 µmol m$^{-2}$ day$^{-1}$
corresponding to 14 - 24% of the DMS flux (MeSH/(MeSH+DMS)). Spearman rank correlations with ocean
biogeochemical parameters showed a moderate to strong positive and highly significant relationship between both
$MeSH_a$ and $DMS_a$ with seawater DMS ($DMS_{sw}$), and a moderate correlation with total dimethylsulfoniopropionate
(total DMSP). A positive correlation of $acetone_a$ with water temperature and negative correlation with nutrient
concentrations is consistent with reports of acetone production in warmer subtropical waters. Positive correlations
of $acetone_a$ with cryptophyte and eukaryotic phytoplankton numbers, and high molecular weight sugars and
Chromophoric Dissolved Organic Matter (CDOM), suggest an organic source. This work points to a significant
ocean source of MeSH, highlighting the need for further studies into the distribution and fate of MeSH, and
suggests links between atmospheric acetone levels and biogeochemistry over the mid-latitude ocean.
In addition, an intercalibration of $DMS_a$ at ambient levels using three independently calibrated instruments showed
~15-25% higher mixing ratios from an Atmospheric Pressure Ionisation-Chemical Ionisation Mass Spectrometer
(mesoCIMS) compared to a Gas Chromatograph with Sulfur Chemiluminescence Detector (GC-SCD) and proton
transfer reaction mass spectrometer (PTR-MS). PTR-MS and mesoCIMS showed similar temporal behaviour with
differences in ambient mixing ratios likely influenced by the $DMS_a$ gradient above the sea surface.
## 1 Introduction
Volatile organic compounds (VOC) are ubiquitous in the atmosphere and have a central role in secondary particle
and tropospheric ozone formation, as well as controlling the oxidative capacity of the atmosphere. VOCs may
also impact air quality and human health, through their role in particle and ozone formation, and direct impacts



through exposure. The role of the ocean in the global cycle of several VOCs is becoming increasingly recognised,
with recent studies showing that the ocean serves as a major source, sink, or both for many pervasive and climate-
active VOCs (Law et al., 2013; Liss and Johnson, 2014; Carpenter and Nightingale, 2015).
The ocean is a major source of reduced volatile sulfur gases (Lee and Brimblecombe, 2016) and the most well-
studied of these is dimethyl sulfide (DMS). Since the publication of the CLAW hypothesis (Charlson et al., 1987),
extensive investigations have been undertaken into DMS formation and destruction pathways, ocean-atmosphere
transfer, and atmospheric transformation and impacts on chemistry and climate (Law et al., 2013; Liss and
Johnson, 2014; Carpenter et al., 2012; Quinn and Bates, 2011). Methanethiol or methyl mercaptan (MeSH) is
another reduced volatile organic sulfur gas which originates in the ocean, with a global ocean source estimated to
be ~17% of the DMS source. The MeSH ocean source is twice as large as the total of all anthropogenic sources
(Lee and Brimblecombe, 2016). However, the importance of ocean derived MeSH as a source of sulfur to the
atmosphere, and the impact of MeSH and its oxidation products on atmospheric chemistry and climate has been
little-studied.
DMS and MeSH in seawater ($DMS_{sw}$ and $MeSH_{sw}$) are both produced from precursor dimethylsulfoniopropionate
(DMSP), which is biosynthesised by different taxa of phytoplankton and released into seawater as a result of
aging, grazing, or viral attack (Yoch, 2002). DMSP is then degraded by bacterial catabolism (enzyme catalysed
reaction) via competing pathways that produce either DMS or MeSH (Yoch, 2002). Recent research showed that
bacterium *Pelagibacter* can simultaneously catabolise both $DMS_{sw}$ and $MeSH_{sw}$ (Sun et al., 2016), although it is
not known how widespread this phenomenon is. DMS may also be produced by phytoplankton that directly cleave
DMSP into DMS (Alcolombri et al., 2015).   Once released, $MeSH_{sw}$ and $DMS_{sw}$ undergo further reaction in
seawater.  These compounds may be assimilated by bacteria, converted to dissolved non-volatile sulfur, be
photochemically destroyed, or in the case of $MeSH_{sw}$, react with dissolved organic matter (DOM) (Kiene and
Linn, 2000; Kiene et al., 2000; Flöck and Andreae, 1996).  $MeSH_{sw}$ has a much higher loss rate constant than
$DMS_{sw}$, with a lifetime on the order of minutes to an hour, compared to ~ days for $DMS_{sw}$ (Kiene, 1996; Kiene
and Linn, 2000). A fraction (~10%) of $DMS_{sw}$ ventilates to atmosphere where it can influence particle numbers
and properties through its oxidation products (Simó and Pedrós-Alió, 1999; Malin, 1997). The fraction of $MeSH_{sw}$
ventilating to the atmosphere is poorly constrained.

While $DMS_{sw}$ measurements are relatively widespread, only a few studies have measured $MeSH_{sw}$. During an
Atlantic Meridional Transect cruise in 1998 (Kettle et al., 2001) $MeSH_{sw}$ was higher in coastal and upwelling
regions with the ratio of $DMS_{sw}$ to $MeSH_{sw}$ varying from unity to 30.  Leck et al (1991) also reported ratios of
$DMS_{sw}/MeSH_{sw}$ of 16, 20 and 6 in the Baltic, Kattegat/Skagerrak and North Seas respectively.  The drivers of this
variability are unknown, but likely due to variation in the dominant bacterial pathway and/or spatial differences
in degradation processes. More recent $MeSH_{sw}$ measurements in the subarctic NE Pacific Ocean showed the ratio
of $DMS_{sw}/MeSH_{sw}$ varied from 2-5 indicating that $MeSH_{sw}$ was a significant contributor to the volatile sulfur pool
in this region (Kiene et al., 2017).  $MeSH_{sw}$ measurements from these three studies (Kettle et al., 2001; Leck and
Rodhe, 1991; Kiene et al., 2017) were also used to calculate the ocean-atmosphere flux of MeSH, assuming control
from the water side. The flux of MeSH/(MeSH+DMS) ranged from 4-5% in the Baltic and Kattegat sea and 11%
in the North Sea (Leck and Rodhe, 1991), 16% over the North/South Atlantic transect (Kettle et al., 2001), and



~15% over the North East Sub-arctic Pacific (Kiene et al., 2017). In a review of global organosulfide fluxes, Lee
and Brimblecombe (2016) estimated that ocean sources provide over half of the total global flux of MeSH to the
atmosphere, with a total 4.7 Tg S a$^{-1}$, however this estimate is based on a voyage-average value from a single
study (Kettle et al., 2001) in which flux measurements varied by several orders of magnitude.
There are very few published atmospheric measurements of MeSH$_a$ over the ocean. To the best of our knowledge,
the only prior MeSH$_a$ measurements over the ocean were made in 1986 over the Drake Passage and the coastal
and inshore waters west of the Antarctic Peninsula (Berresheim, 1987). MeSH$_a$ was detected occasionally at up
to 3.6 ppt, which was roughly 3% of the measured atmospheric DMS$_a$ levels (Berresheim, 1987).
Once MeSH$_{sw}$ is transferred from ocean to atmosphere (MeSH$_a$), the main loss pathway for MeSH$_a$ is via reaction
with OH and NO$_3$ radicals. MeSH$_a$ reacts with OH at a rate 2-3 times faster than DMS, and as such MeSH$_a$ has
an atmospheric lifetime of only a few hours (Lee and Brimblecombe, 2016). The oxidation pathways and products
that result from MeSH$_a$ degradation are still highly uncertain (Lee and Brimblecombe, 2016; Tyndall and
Ravishankara, 1991), though may be somewhat similar to DMS (Lee and Brimblecombe, 2016). This leads to
uncertainty around the final atmospheric fate of the sulfur emitted via MeSH and also the overall impact of MeSH$_a$
oxidation on atmospheric chemistry, particularly in regions when MeSH is a significant proportion of total sulfur
emitted.
For oxygenated VOCs (OVOCs), whether the ocean acts as a source or a sink in a particular region depends on
the concentration gradient between seawater and atmosphere (Carpenter et al., 2012). In the case of acetone,
positive fluxes from the ocean have been observed in biologically productive areas (Taddei et al., 2009) and over
some subtropical ocean regions (Beale et al., 2013; Yang et al., 2014a; Tanimoto et al., 2014; Schlundt et al,
2017), however in other subtropical regions, and generally in oligotrophic waters and at higher latitudes, net fluxes
are zero (e.g. ocean and atmosphere in equilibrium), or negative (transfer of acetone into ocean) (Yang et al.,
2014a; Marandino et al., 2005; Beale et al., 2015; Yang et al., 2014b; Schlundt et al., 2017). Atmospheric acetone
(acetone$_a$) also has significant terrestrial sources including direct biogenic emissions from vegetation, oxidation
of anthropogenic and biogenic hydrocarbons, (predominantly alkanes) and biomass burning (Fischer et al., 2012).
In the ocean, acetone$_{sw}$ is produced photochemically from Chromophoric Dissolved Organic Matter (CDOM),
either directly by direct photolysis or via photosensitizer reactions (Zhou and Mopper, 1997; Dixon et al., 2013;
de Bruyn et al., 2012; Kieber et al., 1990). There is also evidence of direct biological production by marine bacteria
(Nemecek-Marshall et al., 1995) and phytoplankton (Schlundt et al., 2017; Sinha et al., 2007; Halsey et al., 2017).
Furthermore, acetone$_{sw}$ has been found to decrease with depth (Beale et al., 2015; Yang et al., 2014a; Beale et al.,
2013; Williams et al., 2004), pointing to the importance of photochemistry and/or biological activity as the source.
Studies have shown acetone$_{sw}$ production linked to photosynthetically active radiation (PAR) and net shortwave
radiation (Sinha et al., 2007; Beale et al., 2015; Zhou and Mopper, 1997), and Beale et al (2015) found higher
acetone$_{sw}$ concentrations in spring and summer compared to autumn and winter. Removal processes include
uptake of acetone by bacteria as a carbon source (Beale et al., 2013; Halsey et al., 2017; Beale et al., 2015; Dixon
et al., 2013), gas transfer into the atmosphere, vertical mixing into the deep ocean, and photochemical destruction
(Carpenter and Nightingale, 2015).





There are relatively few observations of acetone$_{sw}$ and acetone$_a$ over the remote ocean, particularly in mid and
high latitude regions. An understanding of the spatial distribution of acetone is particularly important due to the
high degree of regional variation in the direction and magnitude of the acetone flux.
In this work, DMS$_a$, MeSH$_a$ and acetone$_a$ measurements were made over a biologically productive region of the
remote South West Pacific Ocean. The relationships between atmospheric levels of these gases are explored, as
well as the relationship with ocean biogeochemical parameters. The importance of MeSH as a source of sulfur to
the atmosphere in this region is estimated and compared to other studies. Finally, we present results from a DMS$_a$
method comparison which was undertaken at sea between three independently calibrated measurement
techniques.
**2 Method**
**2.1 Voyage**
The Surface Ocean Aerosol Production (SOAP) voyage took place on the NIWA RV *Tangaroa* over the
biologically productive frontal waters of Chatham Rise (44°S, 174–181°E), east of New Zealand in the South West
Pacific Ocean. The 23 day voyage took place during the austral summer in February – March 2012. The scientific
aim was to investigate interactions between the ocean and atmosphere, and as such the measurement program
included comprehensive characterisation of ocean biogeochemistry, measurement of ocean-atmosphere gas and
particle fluxes and measurement of trace gases and aerosols distribution and composition in the marine boundary
layer (MBL) (Law et al., 2017). During the voyage, NASA MODIS ocean colour images and underway sensors
were used to identify and map phytoplankton blooms. Three blooms were intensively targeted for measurement:
1) a dinoflagellate bloom with elevated Chl $a$, DMS$_{sw}$ and pCO$_2$ drawdown and high irradiance (bloom 1-B1), 2)
a coccolithophore bloom (bloom 2 – B2) and 3) a mixed community bloom of coccolithophores, flagellates and
dinoflagellates sampled before (bloom 3a –B3a) and after (bloom 3b – B3b) a storm. For further voyage and
measurement details see Law et al., (2017).
**2.2 PTR-MS**
A high sensitivity proton transfer reaction mass spectrometer (PTR-MS) (Ionicon Analytik) was used to measure
DMS, acetone and methanethiol. The PTR-MS sampled from a 25m 3/8 inch ID PFA inlet line which drew air
from the crow's nest of the vessel, 28 m above sea level (a.s.l) at 10 L min$^{-1}$. A baseline switch based on relative
wind speed and direction was employed to minimise flow of ship exhaust down the inlet (see Lawson et al., 2015).

PTR-MS instrument parameters were as follows: inlet and drift tube temperature of 60°C, a 600V drift tube and
2.2 mbar drift tube pressure (133 Td). The O$_2$ signal was < 1% of the primary ion H$_3$O$^+$ signal. DMS, acetone and
MeSH were measured at m/z 63, 59 and 49 respectively with a dwell time of 10s. From day of year (DOY) 43 –
49, 19 selected ions including m/z 59 and m/z 63 were measured resulting in 17 mass scans per hour, however
from DOY 49 the PTR-MS measured in scan mode from m/z 21–155, allowing three full mass scans per hour. As
such, MeSH measurements (m/z 49) were made only from DOY 49 onward.



VOC-free air was generated using a platinum-coated glass wool catalyst heated to 350°C; 4 times per day this air
was used to measure the background signal resulting from interference ions and outgassing of materials.  An
interpolated background signal was used for background correction. Calibrations of DMS and acetone were
carried out daily by diluting calibration gas into VOC – free ambient air (Galbally et al. 2007). Calibration gases
used were a custom ~1 ppm VOC mixture in nitrogen containing DMS and acetone (Scott Specialty gases) and a
custom ~1 ppm VOC calibration mixture in nitrogen containing acetone (Apel Riemer). The calibration gas
accuracy was ± 5%. A calibration gas for MeSH was not available during this voyage. The instrument response
factor for DMS at m/z 63 was also applied to MeSH at m/z 49.  DMS and MeSH have similar collision rate
constants (Williams et al., 1998) and m/z 63 and m/z 49 had the same transmission efficiency. The instrument
response to DMS and acetone varied by 2% and 5% throughout the voyage respectively.
In this work m/z 59 is assumed to be dominated by acetone. Propanal could also contribute to m/z 59, although
studies suggest this is likely low (Beale et al., 2013; Yang et al., 2014a). Similarly, m/z 49 has been attributed to
methanethiol, based on a literature review (Feilberg et al., 2010; Sun et al., 2016), and a lack of likely other
contributing species at m/z 49 in the MBL. As such m/z 59 and m/z 49 represent an upper limit for acetone and
MeSH respectively.
The minimum detectable limit (MDL) for a single 10 s measurement of a selected mass was determined using the
principles of ISO 6879 (ISO, 1995).  Average detection limits for the entire voyage were as follows: m/z 59
(acetone) 24 ppt, m/z 63 (DMS) 22 ppt, m/z 49 (MeSH) 10 ppt.  The percentage of 10 s observations above
detection limits were as follows - m/z 59 100%; m/z 63 98%; and m/z 49 63%.  Inlet losses were determined to
be < 2% for isoprene, monoterpenes, methanol and dimethyl sulfide. Acetone and MeSH losses were not
determined.
**2.2 DMS Intercomparison**
During the SOAP voyage DMS$_a$ measurements were made using three independently calibrated instruments;
Atmospheric Pressure Ionisation-Chemical Ionisation Mass Spectrometer (mesoCIMS) from the University of
California Irvine (UCI), (Bell et al., 2013, 2015), an Ionicon PTR-MS operated by CSIRO (Lawson et al., 2015),
and a HP Gas Chromatograph with Sulfur Chemiluminescence Detector (GC-SCD) operated by NIWA (Walker
et al., 2016).

Details of the mesoCIMS and GC-SCD measurement systems are provided by Bell et al. (2015) and Walker et al.
(2016) with a brief description provided here. The mesoCIMS instrument (Bell et al., 2013) ionizes DMS to DMS-
H+; m/z=63) by atmospheric pressure proton transfer from $H_3O^+$ by passing a heated air stream over a radioactive
nickel foil (Ni-63). The mesoCIMS drew air from the eddy covariance set up on the bow mast at approximately
12m a.s.l.  The inlet was a 1/2" ID PFA tube with a total inlet length of 19m and a turbulent flow at 90 SLPM.
The mesoCIMS sub-sampled from the inlet at 1 L m$^{-1}$. A gaseous tri-deuterated DMS standard (D3-DMS) was
added to the air sample stream at the entrance to the inlet. The internal standard was ionized and monitored
continuously in the mass spectrometer at m/z=66, and the atmospheric DMS mixing ratio was computed from the



measured 63/66 ratio. The internal standard was delivered from a high pressure aluminium cylinder and calibrated
against a DMS permeation tube prior to and after the cruise (Bell et al., 2015).
The GC-SCD system included a semi-automated purge and trap system, a HP 6850 gas chromatograph with
cryogenic preconcentrator/thermal desorber and sulfur chemiluminescence detection (Walker et al 2016). The
system was employed during the voyage for discrete DMS seawater measurements and gradient flux measurement
bag samples (Smith et al., 2018). The system was calibrated using an internal methylethylsulfide (MES)
permeation tube and external DMS permeation tube located in a Dynacalibrator with a twice daily 5-point
calibration and a running standard every 12 samples (Walker et al., 2016).
A DMS measurement intercomparison between the mesoCIMS, GC-SCD and PTR-MS was performed during the
voyage on DOY 64 and DOY 65. Tedlar bags (70 L) with blackout polythene covers were filled with air containing
DMS at sub-ppb levels and were sequentially distributed between all instruments for analysis within a few hours.
On DOY 64, two bags were prepared including ambient air filled from the foredeck and a DMS standard prepared
using a permeation device (Dynacalibrator) and dried compressed air (DMS range 384 – 420 ppt from permeation
uncertainty). On DOY 65, two additional bags were prepared including one ambient air from the foredeck with
tri-deuterated DMS added and a DMS standard prepared using the Dynacalibrator and dried compressed air (DMS
range 331 – 363 ppt). MesoCIMs values are not available for DOY 64 due to pressure differences between bag
and instrument calibration measurements; this was resolved by using an internal standard on DOY 65. For those
analyses, the mesoCIMS and PTR-MS measured DMS at m/z 63 and tri-deuterated DMS at m/z 66, while the
GC-SCD measured both DMS and deuterated DMS as a single peak.
**2.4 Biogeochemical measurements in surface waters**
Continuous seawater measurements were obtained from surface water sampled by an intake in the vessel's bow
at a depth of ~7m during the SOAP voyage and included underway temperature and salinity (Seabird
thermosalinograph SBE-21), underway chlorophyll $a$ (Chl a) and backscatter (Wetlabs (Seabird) ECOtriplet),
pCO2 (Currie et al., 2011), dissolved DMS ($DMS_{sw}$) (miniCIMS) (Bell et al., 2015). Quenching obscured the Chl
a signal during daylight when irradiance > 50 W m$^{-2}$.

The following parameters were measured in surface waters (depths 2-10 m) in discrete samples from Niskin
bottles on a CTD rosette: nutrients according to methods described in Law et al., (2011), particulate nitrogen
concentration (Nodder et al., 2016), phytoplankton speciation, groups and numbers (optical microscopy of
samples preserved in Lugol's solution) (Safi et al., 2007), Flow cytometry, (Hall and Safi, 2001). In addition,
organic parameters measured included High Molecular Weight reducing sugars (Somogyi, 1926, 1952; for details
see Burrell (2015)) and CDOM measured using a Liquid Waveguide Capillary Cell (Gall et al., 2013). See Law
et al., (2017) for further details and results for these parameters.
**3 Results and discussion**
**3.1 DMS atmospheric intercomparison**
This section describes a comparison of $DMS_a$ measurements from bag samples of ambient air and DMS standard
mixtures (analysed by GC-SCD, PTR-MS and mesoCIMS, see Section 2), as well as comparison of ambient $DMS_a$
measurements (PTR-MS and mesoCIMS).
**Comparison of bag samples**
Table 1 summarises the comparison between the GC-SCD, PTR-MS and mesoCIMS instruments for ambient and
DMS standard bags prepared and analysed on DOY 64 and 65 (see Section 2.2). The highest DMS levels were
measured by the mesoCIMS with GC-SCD and PTR-MS ~20-25 % and ~20-30% lower respectively. The GC-
SCD and PTR-MS agreed reasonably well, with a mean difference of 5% (range 0-10%) between instruments for
different diluted standard and ambient air bags. There was no clear influence of dry versus humid (ambient) bag
samples on the differences between instruments.
**Comparison of in situ ambient measurements**
Measurements from the PTR-MS and mesoCIMS were interpolated to a common time stamp for comparison and
differences examined only where data were available for both instruments. PTR-MS results for DMS were
reported for 10 s every 4 minutes until DOY 49 and then 10 s every 20 minutes until the end of the voyage (Section
2.2). The mesoCIMS measured DMS continuously and reported 10 minute averages. As such the PTR-MS
measured only a 'snapshot' of the $DMS_a$ levels in each measurement cycle of 4 or 20 minutes. This was a potential
source of difference between the two instruments when DMS levels changed rapidly (Bell et al., 2015).
The PTR-MS and mesoCIMS drew air from separate intakes, with heights of 28 m and 12 m a.s.l, respectively.
As such, a further source of the difference between the PTR-MS and mesoCIMS measurements is likely due to
vertical gradients in DMS caused by turbulent mixing of the local surface DMS flux into the atmospheric surface
layer. On days with a strong DMS source and/or more stable stratification in the boundary layer, a significant
decrease with height is expected (Smith et al., 2018). If all the DMS observed was due to local emissions, the
vertical gradient would be described by Equation 2 from Smith et al (2018):

$$F \equiv -u*C* = -\frac{u*k}{\varphi c\,(z/L)}\left(\frac{\partial c}{\partial lnz}\right)$$   (1)

Where $u*$ is friction velocity, C* is scaling parameter for gas concentration, $k$ is the von Kármán constant, $\varphi c$ is
the stability function for mass, $z$ is the height above mean water level and L is the Monin-Obukhov scaling length
representing atmospheric stability.

Figure 1 shows wind speed, absolute wind direction and atmospheric stability, $DMS_a$ levels from the voyage
measured by PTR-MS and mesoCIMS, relative percent difference between the two measurements (normalised to
the mesoCIMS), and observed absolute difference in $DMS_a$ between the two measurements, as well as the
expected calculated difference (Eq 1) between two measurements due to the $DMS_a$ concentration gradient.





The mesoCIMS and PTR-MS $DMS_a$ data showed similar temporal behaviour over the voyage (Fig. 1). From DOY
44 – 46 there was an average of 50% (±10%) relative difference between measurements, yet on DOY 47 this
difference decreased suddenly to an average of ~20% (±20%). The reason for this change at DOY 47 is unknown.
Overall, agreement between instruments improved with time during the voyage, with differences of several
hundred ppt of DMS observed in the first few days decreasing to differences of only 10-20 ppt by the end of the
voyage. The agreement between instruments improves with increasing wind speeds (Fig. 1). The expected
calculated difference between $DMS_a$ at the two inlet heights due to the DMS concentration gradient also decreases
throughout the voyage. This indicates that the increasing agreement between instruments during the voyage was
likely influenced by a progressively well mixed atmosphere leading to weaker DMS vertical gradients. Prior to
DOY 47 the difference between PTR-MS and mesoCIMS appears to have been due to instrument calibration or
other instrument differences rather than the DMS concentration gradient.
Figure 2a shows paired $DMS_a$ data from the mesoCIMS versus PTR-MS over the whole voyage and Fig 2b shows
paired mesoCIMS data versus PTR-MS data converted to same height as the mesoCIMS with the expected DMS
difference calculated from the eddy covariance estimate of DMS flux (from mesoCIMS) and eddy diffusivity
(PTR-MS $DMS_a$ + calculated difference between the two intake heights). The reduced major axis regression
relationship between the two measurements systems for uncorrected data gives a slope of $0.74 \pm 0.02$, while for
the corrected data gives $0.81 \pm 0.02$. The gradient-corrected slope agrees with the ambient bag sample ratio from
the method comparison (PTR-MS / mesoCIMS $=0.81 \pm 0.16$) (Table 1). Correcting for the DMS gradient
improved the comparison between PTR-MS and mesoCIMS. The remaining ~20% difference is likely due to
instrument calibration differences and differing approaches of integrated versus discrete measurements.
There was no obvious impact of absolute wind direction on the differences observed between measurement
systems. Note that due to the Baseline switch which was employed to avoid sampling ship exhaust down the PTR-
MS inlet (Lawson et al., 2015) the PTR-MS did not sample during certain relative wind directions. However, this
does not affect the comparison which was undertaken only when data were available for both instruments.

### 3.2 Ambient atmospheric data

Atmospheric mixing ratios of $MeSH_a$, $DMS_a$ and $acetone_a$ are shown along the voyage track in Fig. 3 with bloom
locations highlighted. Figure 4 shows a time series of $MeSH_a$, $DMS_a$, $acetone_a$, $MeSH_a/DMS_a$ (all measured with
PTR-MS) as well as $DMS_{sw}$ (miniCIMS) from Bell et al (2015), Chl$a$, irradiance, wind speed, wind direction and
sea and air temperature. Note that $MeSH_a$ measurements started on DOY 49, the last day of bloom B1. The fraction
of back trajectories arriving at the ship that had been in contact with land masses in the previous 10 days is also
shown with a value of 0 indicating no contact with land masses in the preceding 10 days. This was calculated
using the Lagrangian Numerical Atmospheric-dispersion Modelling Environment (NAME) for the lower
atmosphere (0–100 m) as time-integrated particle density (g s $m^{-3}$), every 3 hours from ship location (Jones et al,
2007) as shown in Law et al. (2017). Where air contacted land masses this was the New Zealand land mass in
almost all cases.



MeSH$_a$ ranged from below detection limit (< 10 ppt) to 65 ppt, DMS$_a$ ranged from below detection limit (~22 ppt)
up to 957 ppt, and acetone$_a$ ranged from 50-1500 ppt (Table 2). The ratio of MeSH$_a$ to DMS$_a$ ranged from 0.03 -
0.36 for measurements when both were above the MDL. Periods of elevated DMS$_a$ generally correspond to periods
of elevated DMS$_{sw}$. Both DMS$_a$ and DMS$_{sw}$ were very high during B1, during the transect to B2, and the first half
of B2 occupation. MeSH$_a$ variability broadly correlates with DMS$_a$ and DMS$_{sw}$, with highest levels during B2
(no data available for B1). The highest acetone$_a$ levels observed occur during B2, and a broad acetone peak during
B1 of 700 ppt (~DOY 49) overlaps with but is slightly offset from the largest DMS$_a$ peak during the voyage
(~957 ppt). DMS$_a$, acetone$_a$ and MeSH$_a$ were somewhat lower during B3a and lowest during the B3b, the post-
storm part of that bloom B3 (see Law et al., 2017). In general, DMS$_a$ levels during B1 were at the upper range of
those found in prior studies elsewhere (Lana et al., 2011;Law et al., 2017). MeSH$_a$ levels during B1 were
substantially higher than the only comparable measurements from the Drake Passage and the coastal and inshore
waters west of the Antarctic Peninsula (3.6 ppt) (Berresheim, 1987). The average acetone$_a$ levels during this study
were broadly comparable to those from similar latitudes reported in the South Atlantic and Southern Ocean
(Williams et al., 2010) and at Cape Grim (Galbally et al., 2007). Acetone$_a$ during SOAP was generally lower than
at similar latitudes at Mace Head (Lewis et al., 2005), the Southern Indian Ocean (Colomb et al., 2009) and also
the marine subtopics (Read et al., 2012; Schlundt et al., 2017; Warneke and de Gouw, 2001; Williams et al., 2004).
There were two occasions when elevated acetone$_a$ corresponded closely to increased land influence – during B1
on DOY 48 - 49 (maximum land influence 12%) and DOY 60 (maximum land influence 20%) (Fig 4). Both these
periods corresponded to winds from the north, and back trajectories show that the land mass contacted was the
southern tip of New Zealand's North Island (including the city of Wellington and the northern section of the South
Island in both cases). The acetone measured during these periods may have been emitted from anthropogenic and
biogenic sources and from photochemical oxidation of hydrocarbon precursors (Fischer et al., 2012). The acetone
enhancement relative to the degree of land influence was higher on DOY 48 – 49 than DOY 60 possibly due to
different degrees of dilution of the terrestrial plume, or different terrestrial source strengths.
The period with the highest acetone levels during B2 (1508 ppt) corresponds with a period of negligible land
influence (0.3%) indicating a non-terrestrial, possibly local source of acetone$_a$. Neither MeSH$_a$ or DMS$_a$ maxima
corresponded with peaks in land influence, except for the latter part of the DMS$_a$ maximum on DOY 48-49;
however the source of DMS$_a$ during DOY 48 – 49 is attributed to local ocean emissions as shown by strong
association between DMS$_{sw}$ and DMS$_a$ during this period (Fig. 4).

Correlations of DMS$_a$, MeSH$_a$ and acetone$_a$ were examined to identify possible common marine sources or
processes influencing atmospheric levels (Table 3). Only data above MDL were included in the regressions.
Acetone$_a$ data likely influenced by terrestrial sources (DOY 48-49 and 60, described above) were removed from
this analysis. A moderate correlation ($R^2$=0.5, p<0.0001) was found between DMS$_a$ and MeSH$_a$ during B2 with
a correlation of $R^2$=0.3, (p<0.0001) between DMS$_a$ and MeSH$_a$ for all data (Fig. 5). During B2 the slope was 0.13
(MeSH$_a$ roughly 13% of the DMS$_a$ mixing ratios), while for all data the slope was 0.07 (including blooms and
transiting between blooms).





MeSH$_{sw}$ and DMS$_{sw}$ are produced from bacterial catabolism of DMSP via two competing processes, so the amount
of DMS$_{sw}$ vs MeSH$_{sw}$ produced from DMSP will depend on the relative importance of these two pathways at any
given time. Additional sources of DMS$_{sw}$, such as phytoplankton that cleave DMSP into DMS will also influence
the amount of DMS$_{sw}$ vs MeSH$_{sw}$ produced.  A phytoplankton-mediated source of DMS$_{sw}$ was likely to be an
important contributor to the DMS$_{sw}$ pool during the SOAP voyage, either through indirect processes (zooplankton
grazing, viral lysis and senescence) or direct processes (algal DMSP-lyase activity) (Lizotte et al., 2017). The
relative loss rates of DMS$_{sw}$ and MeSH$_{sw}$ through oxidation, bacterial uptake or reaction with DOM will also
influence the amount of each gas available to transfer to the atmosphere, with MeSH$_{sw}$ having a much faster loss
rate in seawater than DMS$_{sw}$ (Kiene and Linn, 2000; Kiene et al., 2000).  Differences between the gas transfer
velocities of DMS and MeSH would also affect the atmospheric mixing ratios.  Such differences are likely to be
small, due to similar solubilities (Sander, 2015) and diffusivities (Johnson, 2010) (see Section 3.4). A final factor
that will influence the slope of DMS$_a$ vs MeSH$_a$ is the atmospheric lifetime (Table 2). The average lifetimes of
DMS$_a$ and MeSH$_a$ in this study are estimated at 24 and 9 hours respectively with respect to OH, calculated using
DMS reaction rate of OH from Berresheim et al. (1987), the MeSH reaction rate from Atkinson et al. (1997) and
OH concentration calculated as described in Lawson et al. (2015).  Hence, the correlation between DMS$_a$ and
MeSH$_a$ reflects the common seawater source of both gases, while the differing slopes between B2 and all data
probably reflect the different sources and atmospheric lifetimes.  While a correlation between MeSH and DMS
has been observed in seawater samples previously (Kettle et al., 2001; Kiene et al., 2017), to our knowledge this
is the first time that a correlation between MeSH$_a$ and DMS$_a$ has been observed in the atmosphere over the remote
ocean.
There were several weak (R$^2$ ≤ 0.2) but significant correlations between DMS$_a$ and acetone$_a$, and acetone$_a$ and
MeSH$_a$ (Table 3).  The correlation of acetone$_a$ with DMS$_a$ may reflect elevated organic sources for photochemical
production of acetone in regions of high dissolved sulfur species. A further discussion of drivers of DMS$_a$, acetone$_a$
and MeSH$_a$ mixing ratios is provided in Section 3.3.
Figure 6 shows the voyage-average diurnal cycles for DMS$_a$, MeSH$_a$ and acetone$_a$. The diurnal cycle of DMS$_a$
shows variations by almost a factor of 3 from morning (maximum  at 8:00 hrs ~ 330 ppt) to late afternoon
(minimum, 16:00 hrs ~ 120 ppt).  A DMS$_a$ diurnal cycle with sunrise maximum and late afternoon minimum has
been observed in many previous studies and is attributed to photochemical destruction by OH. This includes Cape
Grim baseline station which samples air from the Southern Ocean (average minimum and maximum ~40-70 ppt)
(Ayers and Gillett, 2000), over the tropical Indian ocean (average minimum and maximum ~25-60 ppt (Warneke
and de Gouw, 2001) and at Kiritimati in the tropical Pacific (average minimum and maximm 120-200 ppt) (Bandy
et al., 1996).  The higher atmospheric levels in this study are due to high DMS$_{sw}$ concentrations (>15 nM). The
amplitude of the DMS diurnal cycle is likely to have been influenced by stationing the vessel over blooms with
high DMS$_{sw}$ from 8:00 hrs each day and regional mapping of areas with lower DMS$_{sw}$ overnight (Law et al., 2017).

The diurnal cycle for MeSH$_a$ (Fig. 6 b) shows similar behaviour to DMS$_a$ with the mixing ratios varying by a
factor of ~2 with the minimum mixing ratio occurring at around 16:00 hrs (the same time as minimum DMS$_a$).





The most important sink of MeSH$_a$ is thought to be oxidation by OH (Lee and Brimblecombe, 2016), and the
minima in late afternoon may be due to destruction by OH.
The acetone$_a$ diurnal cycle (Fig. 6c) with land-influenced data removed shows reasonably consistent mixing ratios
from the early morning until midday, with an overall increase in acetone levels during the afternoon hours from
14:00 hrs onwards, then decreasing again at night, which is the opposite to the behaviour of DMS$_a$ and MeSH$_a$.
Acetone is long lived (~60 days – Table 2) with respect to oxidation by OH. The increase of acetone$_a$ mixing
ratios in the afternoon may indicate photochemical production from atmosphere or sea surface precursors but there
was no correlation between irradiance and acetone$_a$ during the voyage.
**3.3 Flux calculation from nocturnal accumulation of MeSH**
MeSH and DMS fluxes ($F$) were calculated according to the nocturnal accumulation method (Marandino et al,
2007). This approach assumes that nighttime photochemical losses are negligible, and that sea surface emissions
accumulate overnight within the well-mixed marine boundary layer (MBL). Horizontal homogeneity and zero
flux at the top of the boundary layer are also assumed. The air-sea flux is calculated from the increase in MeSH
and DMS. For example:
$$F = \frac{\partial [MeSH]}{\partial t} \times h \qquad\qquad (2)$$
where [MeSH] is the concentration of MeSH in mol m$^{-3}$ and $h$ = average nocturnal MBL for the voyage of 1135
m ± 657 m, estimated from nightly radiosonde flights.
DMS and MeSH fluxes were calculated for 3 nights (DOY 52, 54 and 60) (Table 4) when linear increases in
mixing ratios occurred over several hours (Fig 4). The MeSH flux was lowest on DOY 52 prior to B2 (3.5 ± 2
µmol$^{-1}$ m$^{-2}$ day$^{-1}$), higher on DOY 60 during B3a (4.8 ± 2.8 µmol$^{-1}$ m$^{-2}$ day$^{-1}$), and highest on DOY 42 during B2
(5.8 ± 3.4 µmol$^{-1}$ m$^{-2}$ day$^{-1}$). There are no MeSH measurements during B1. The percentage of
MeSH/(DMS+MeSH) emitted varied from 14% for DOY 60 (B3a), up to 23% and 24% for DOY 54 (B2) and
DOY 52 (prior to B2).
For comparison the DMS fluxes measured using eddy covariance (EC) at the same time are given in Table 4 (Bell
et al., 2015). DMS fluxes calculated using the nocturnal accumulation method are within the variability of the EC
fluxes (Bell et al., 2015).
The average MeSH flux calculated from this study (4.7 µmol m$^{-2}$ day$^{-1}$) was more than 4 times higher than average
MeSH fluxes from previous studies in the North/South Atlantic (Kettle et al., 2001) and in the Baltic, Kattegat
and North Sea (Leck and Rodhe, 1991) (Table 5). The MeSH fluxes calculated from this work are comparable to
maximum values reported by Kettle et al., (2001) which were observed in localised coastal and upwelling regions.
The average emission of MeSH compared to DMS (MeSH/(DMS+MeSH)) was higher in this study (20%) than
previous studies (Table 5) including the Baltic, Kattegat and North Sea (5%, 4% and 11%), North/South Atlantic
(16%), and a recent study from the Northeast Sub-arctic Pacific (~15%) (Kiene et al., 2017). Note that other
sulfur species such as dimethyl disulphide (DMDS), carbon disulphide (CS$_2$) and hydrogen sulphide (H$_2$S)
typically make a very small contribution to the total sulfur compared to DMS and MeSH (Leck and Rodhe,
1991;Kettle et al., 2001; Yvon et al., 1993) and so are neglected from this calculation.




**3.4 Correlation with ocean biogeochemistry**
To investigate the influence of biogeochemical parameters on atmospheric mixing ratios of $MeSH_a$, $DMS_a$ and
$acetone_a$, Spearman rank correlations were undertaken to identify relationships significant at the 95% confidence
interval (CI). Table 6 summarises the correlation coefficients and p values for significant correlations. $MeSH_a$,
$DMS_a$ and $acetone_a$ data were averaged one hour either side of the CTD water entry time for the analysis.
Sulfur gases $MeSH_a$ and $DMS_a$ are short lived and so the air-sea flux is controlled by the seawater concentration.
By contrast, $acetone_a$ is much longer lived in the atmosphere (~60 days), so the air/sea gradient can be influenced
by both oceanic emissions and atmospheric transport from other sources. As such, the variability in $acetone_a$
mixing ratios may be driven by ocean/air exchange and/or input of $acetone_a$ to the boundary layer from terrestrial
sources, the upper atmosphere, or in situ production. This means that correlation analyses to explore ocean
biogeochemical sources of $acetone_a$ may be confounded by atmospheric sources. Removal of land influenced
data reduces the likelihood of this but observed increases in atmospheric acetone could still be from in situ
processes such as oxidation of organic aerosol or mixing from above the boundary layer.
Both $MeSH_a$ and $DMS_a$ have a strong positive and highly significant relationship with $DMS_{sw}$, and a moderate
correlation with discrete measurements of $DMSP_t$ and $DMSP_p$. The correlation of $DMS_a$ with $DMS_{sw}$ is clear,
however the correlation of $MeSH_a$ with $DMS_{sw}$ is likely due to a common ocean precursor of both gases (DMSP)
albeit via different production pathways. $DMS_a$ and $MeSH_a$ correlate with $DMSP_p$ (particulate) but not with
$DMSP_d$ (dissolved). For $DMS_a$, the correlation may reflect that a proportion of the DMS observed was derived
directly from phytoplankton rather than being bacterially mediated, in agreement with findings by Lizotte et al.,
(2017); however, as demethylation of $DMSP_d$ represents the primary source of MeSH the lack of correlation is
surprising. The latter may reflect MeSH sinks in surface water associated with organics and particles (Kiene,
1996). $DMS_a$ also correlated with particulate nitrogen and showed a moderate negative correlation with silicate
that may reflect lower DMS production in diatom-dominated waters.
$Acetone_a$ shows a positive correlation with temperature and negative correlation with nutrients. This is consistent
with reported sources of $acetone_{sw}$ in warmer subtropical waters (Beale et al., 2013; Yang et al., 2014a; Tanimoto
et al., 2014; Schlundt et al., 2017). The positive relationship with organic material including HMW sugars and
CDOM may reflect a photochemical ocean source (Zhou and Mopper, 1997; Dixon et al., 2013; de Bruyn et al.,
2012; Kieber et al., 1990), or possibly a biological source (Nemecek-Marshall et al., 1995; Nemecek-Marshall et
al., 1999; Schlundt et al., 2017; Sinha et al., 2007; Halsey et al., 2017) as indicated by the correlations with
cryptophyte and picoeukaryote abundance. Correlation with particle backscatter suggests potential links between
$acetone_a$ and coccolithophores (Sinha et al., 2007). Alternatively, the positive correlations of $acetone_a$ with these
organic components of sea water may reflect acetone production in the atmosphere from photochemical oxidation
of ocean-derived organic aerosols (Pan et al., 2009; Kwan et al., 2006; Jacob et al., 2002).



**4 Implications and conclusions**
Mixing ratios of short-lived $MeSH_a$ over the remote ocean of up to 65 ppt in this study provide evidence that
MeSH transfers from the ocean into the atmosphere and may be present at non-negligible levels in the atmosphere
over other regions of high biological productivity. The average MeSH flux calculated from this study (4.7 µmol
$m^{-2}$ $day^{-1}$) was at least 4 times higher than average MeSH fluxes from previous studies and is comparable to
maximum MeSH flux values reported in localised coastal and upwelling regions of the North/South Atlantic
(Kettle et al., 2001) (Table 5). The average emission of MeSH compared to DMS (MeSH/(DMS+MeSH)) was
higher in this study (20%) than previous studies (4-16%), indicating MeSH provides a significant transfer of sulfur
to the atmosphere in this region. Taken together with other studies, the magnitude of the ocean MeSH flux to the
atmosphere appears to be highly variable as is the proportion of S emitted as MeSH compared to DMS. For
example, MeSH fluxes in the Kettle et al. (2001) study varied by orders of magnitude, and in some cases the
MeSH flux equalled the DMS flux. Similarly, studies that reported $MeSH_{sw}$ and $DMS_{sw}$ concentrations have
shown the $DMS_{sw}/MeSH_{sw}$ concentration ratios varied substantially, from 30 to unity (Kettle et al 2001), from 6-
20 (Leck and Rodhe, 1991) and 2-5 (Kiene et al., 2017). As such, further studies are needed to investigate the
spatial distribution of MeSH both in seawater and the atmosphere as well as the importance of MeSH as a source
of atmospheric sulfur. The fate of atmospheric MeSH sulfur in the atmosphere is also highly uncertain, in terms
of its degradation pathways and reactions, and intermediate and final degradation products. For example, the
impact that oxidation of $MeSH_a$ has on the oxidative capacity of the MBL and on other processes such as particle
formation or growth to the best of our knowledge remains largely unknown, and further work is needed on its
atmospheric processes and fate.
This work suggests a source of acetone from warmer subtropical ocean waters, in line with other studies, with
positive correlations between $acetone_a$ and ocean temperature, high molecular weight sugars, cryptophyte and
eukaryote phytoplankton, chromophoric dissolved organic matter (CDOM) and particle backscatter, and a
negative correlation with nutrients. While data with a terrestrial source influence was removed from this analysis,
it is still possible that the acetone peaks observed may not have been due to a positive flux of acetone from the
ocean, but rather from in situ processes leading to acetone production such as oxidation of marine-derived organic
aerosol.
Finally, the SOAP voyage provided the opportunity to compare 3 independently calibrated DMS measurement
techniques at sea (PTR-MS, mesoCIMS and GC-SCD). Agreement was generally good, with a mean difference
of 5% between the PTR-MS and GC-SCD DMS diluted standard and air sample measurements, with the
mesoCIMS mixing ratios approximately 20-30% higher. A comparison of ambient $DMS_a$ data during the voyage
for the PTR-MS and mesoCIMS showed very similar temporal behaviour, and an average difference of ~25%.
Correcting for the expected difference in $DMS_a$ due to the DMS concentration gradient at the different inlet heights
(28 and 12 m a.s.l for the PTR-MS and mesoCIMS respectively) reduced this difference to ~20%. As such, this
remaining difference is likely due to instrument calibration differences and differing approaches of integrated
versus discrete measurements.





**Data availability**
DMS, acetone and MeSH data are available via the CSIRO data access portal (DAP) at
https://doi.org/10.25919/5d914b00c5759. Further data are available by emailing the corresponding author or the
voyage leader: cliff.law@niwa.co.nz.
**Author Acknowledgements**
We thank the officers and crew of the RV Tangaroa and NIWA Vessels for logistics support. Many thanks to John
McGregor (NIWA) for providing land influence data and to Paul Selleck and Erin Dunne (CSIRO) for helpful
discussions. Thanks to the NIWA Visiting Scientist Scheme and CSIRO's Capability Development Fund for
providing financial support for Sarah Lawson's participation in the SOAP voyage.

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





**Table 1. Results of the DMS bag sample intercomparison study undertaken during the SOAP voyage. Note that a 1 s PTR-MS dwell time for m/z 63 and 66 was used during the intercomparison compared to the 10 s during ambient measurements; as such the PTR-MS std dev reported here is expected to be ~3 times higher than during ambient measurements. Total refers to the ambient DMS + spiked tri-deuterated DMS bag sample on DOY 65.**

| DOY | Comparison | DMS (ppt) av ± stdev | | | DMS ratios | | |
| | | GC-SCD | PTR-MS | mesoCIMS | GC-SCD /PTR-MS | PTR-MS /mesoCIMS | GC-SCD /mesoCIMS |
| --- | --- | --- | --- | --- | --- | --- | --- |
| 64 | *Standard (dry)* | $354 \pm 6$ | $339 \pm 64$ | n/a | $1.04 \pm 0.2$ | n/a | n/a |
| 65 | *Standard (dry)* | $289 \pm 2$ | $262 \pm 43$ | $383 \pm 30$ | $1.1 \pm 0.18$ | $0.68 \pm 0.12$ | $0.75 \pm 0.06$ |
| 64 | *Ambient* | $168 \pm 5$ | $158 \pm 49$ | n/a | $1.06 \pm 0.33$ | n/a | n/a |
| 65 | *Ambient* | n/a | $127 \pm 43$ | $141 \pm 5$ | n/a | $0.90 \pm 0.30$ | n/a |
| | *+tri-deuterated DMS* | n/a | $197 \pm 49$ | $260 \pm 2$ | n/a | $0.76 \pm 0.19$ | n/a |
| | *Total* | $323 \pm 9$ | $324 \pm 66$ | $401 \pm 6$ | $1.0 \pm 0.2$ | $0.81 \pm 0.16$ | $0.81 \pm 0.03$ |

**Table 2. MeSH$_a$, DMS$_a$ and acetone$_a$ measured with PTR-MS during the SOAP voyage, reaction rate constant for OH and calculated lifetime with respect to OH**

| | Mean (range) ppt | $k_{OH}$* ($cm^3$ molecule$^{-1}$ s$^{-1}$) | Lifetime (days) |
| --- | --- | --- | --- |
| MeSH | 18 (BDL – 65) | $3.40E^{-11}$ | 0.4 |
| DMS | 208 (BDL – 957) | $1.29E^{-11}$ | 1 |
| acetone | 237 (54-1508) | $2.20E^{-13}$ | 60 |

BDL= below detection limit

*Reaction rate constants from Atkinson 1997 (MeSH), Berresheim et al 1987 (DMS) and Atkinson 1986 (acetone)

**Table 3. Pearson correlations between DMS$_a$ and MeSH$_a$ and acetone$_a$ which are significant at 95 % CI. Land influenced data removed (acetone)**

| | | Slope (p-value) | $R^2$ |
| --- | --- | --- | --- |
| DMS vs MeSH | All data (n=266) | 0.07 (<0.0001) | 0.3 |
| | B2 (n=98) | 0.13 (<0.0001) | 0.5 |
| | B3 (n=76) | 0.03 (0.001) | 0.1 |
| DMS vs acetone | All data (n=1301) | 0.30 (<0.0001) | 0.1 |
| | B1 (n=883) | 0.19 (<0.0001) | 0.1 |
| | B2 (n=122) | 1.1 (<0.0001) | 0.2 |
| Acetone vs MeSH | All data (n=265) | 0.02 (<0.0001) | 0.1 |
| | B3 (n=76) | 0.06 (0.03) | 0.1 |





**Table 4. MeSH and DMS fluxes calculated using the nocturnal buildup method, compared with DMS flux measured**
**using EC method (Bell et al., 2015). The ± values on the MeSH and DMS flux are due to the std deviation of the MBL**
**height.**

| Bloom | DOY | MeSH ppt/hr | DMS ppt/hr | MeSH/ MeSH+DMS (%) | Flux MeSH $\mu mol/m^2/day$ | NBL Flux DMS $\mu mol/m^2/day$ | EC Flux DMS mean ± std dev |
|---|---|---|---|---|---|---|---|
| Just prior to B2 | 52.2 - 52.7 | 3 | 11 | 24 | 3.5 ± 2.0 | 12.7 ± 7.4 | 7.6 ± 4.8 |
| B2 | 54.2 - 54.4 | 5 | 16 | 23 | 5.8 ± 3.4 | 18.5 ± 10.7 | 26.4 ± 9.7 |
| B3a | 60.2- 60.4 | 4 | 27 | 14 | 4.8 ± 2.8 | 31.0 ± 17.9 | 29.4 ± 8.2 |

**Table 5. MeSH flux from this and previous studies (voyage averages)**

| Location | MeSH flux ($\mu mol/m^2/day$) | Flux MeSH/MeSH+DMS (%) | Reference |
|---|---|---|---|
| Baltic sea | 0.2 | 5% | Leck and Rodhe., 1991 |
| Kattegat sea | 0.8 | 4% | |
| North Sea | 1.6 | 11% | |
| North/South Atlantic | 1.2 | 16% | Kettle et al., 2001 |
| Northeast subarctic Pacific | Not reported | ~15% | Kiene et al., 2017 |
| South West Pacific | 4.7 | 20% | This study |

**Table 6.  Spearman rank correlations significant at 95% confidence interval (CI). Correlation coefficient (and p-value)**
**are shown. No entry indicates there was no correlation at 95% CI.**

| | Acetone$_a$ | DMS$_a$ | MeSH$_a$ |
|---|---|---|---|
| **Positive correlations** | | | |
| salinity | 0.55 (0.005) n=25 | | |
| sea temperature | 0.77 (<0.0001) n=25 | | |
| beta -660 backscatter | 0.67 (0.0004) n=25 | | |
| T pCO2 | 0.59 (0.029) n=15 | | |
| DMS$_{sw}$ (nM) | 0.49 (0.025) n=21 | 0.73(0.0002) n=22 | 0.59 (0.011) n=18 |
| Chla/MLD | 0.50 (0.014) n=25 | | |
| particulate nitrogen | | 0.79 (0.048) n=7 | |
| Cryptophyte algae | 0.47 (0.019) n=25 | | |



| Eukaryotic Picoplankton | 0.48 (0.016) n=25 | | |
| DMSPt | | 0.54 (0.011) n=22 | 0.59 (0.014) n=17 |
| DMSPp | | 0.56 (0.007) n=22 | 0.53 (0.032) n=17 |
| CDOM | 0.48 (0.041) n=20 | | |
| HMW reducing sugars | 0.67 (0.011) n=14 | | |
| **Negative correlations** | | | |
| Chl$a$/backscatter 660 | -0.47 (0.019) n=25 | | |
| mixed layer depth | -0.66 (0.0005) n=25 | | |
| dissolved oxygen | -0.45 (0.030) n=24 | | |
| Phosphate | -0.54 (0.006) n=25 | | |
| Nitrate | -0.60 (0.002) n=25 | | |
| Silicate | -0.50 (0.012) n=25 | -0.43 (0.031) n=26 | |
| Monounsaturated fatty acids | -0.82 (0.007) n=10 | | |





2
3

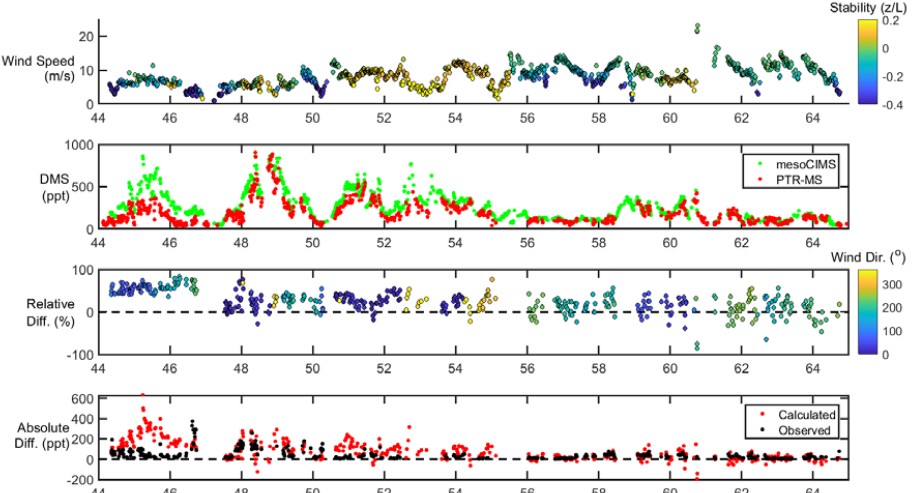

**Figure 1 From top to bottom, wind speed and stability, DMS$_a$ measurements from mesoCIMS and PTR-MS, relative difference (normalised to mesoCIMS) according to absolute wind direction, and absolute observed and calculated difference between mesoCIMS and PTR-MS, taking into account the expected DMS concentration gradient (Eq. 1)**





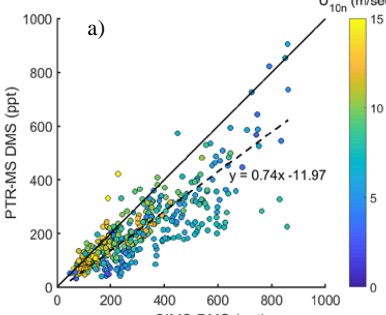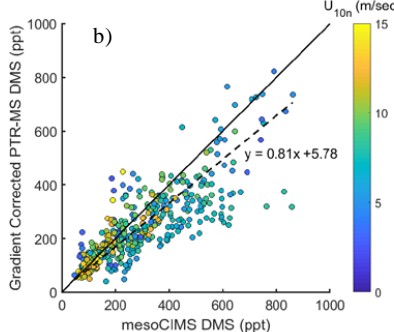

3  **Fig 2 a) DMS$_a$ measured by mesoCIMS (x) and PTR-MS (y) b) mesoCIMS (x) and PTR-MS (y) DMS data corrected**
4  **for the expected concentration gradient (observed PTR-MS DMS + calculated delta DMS)**



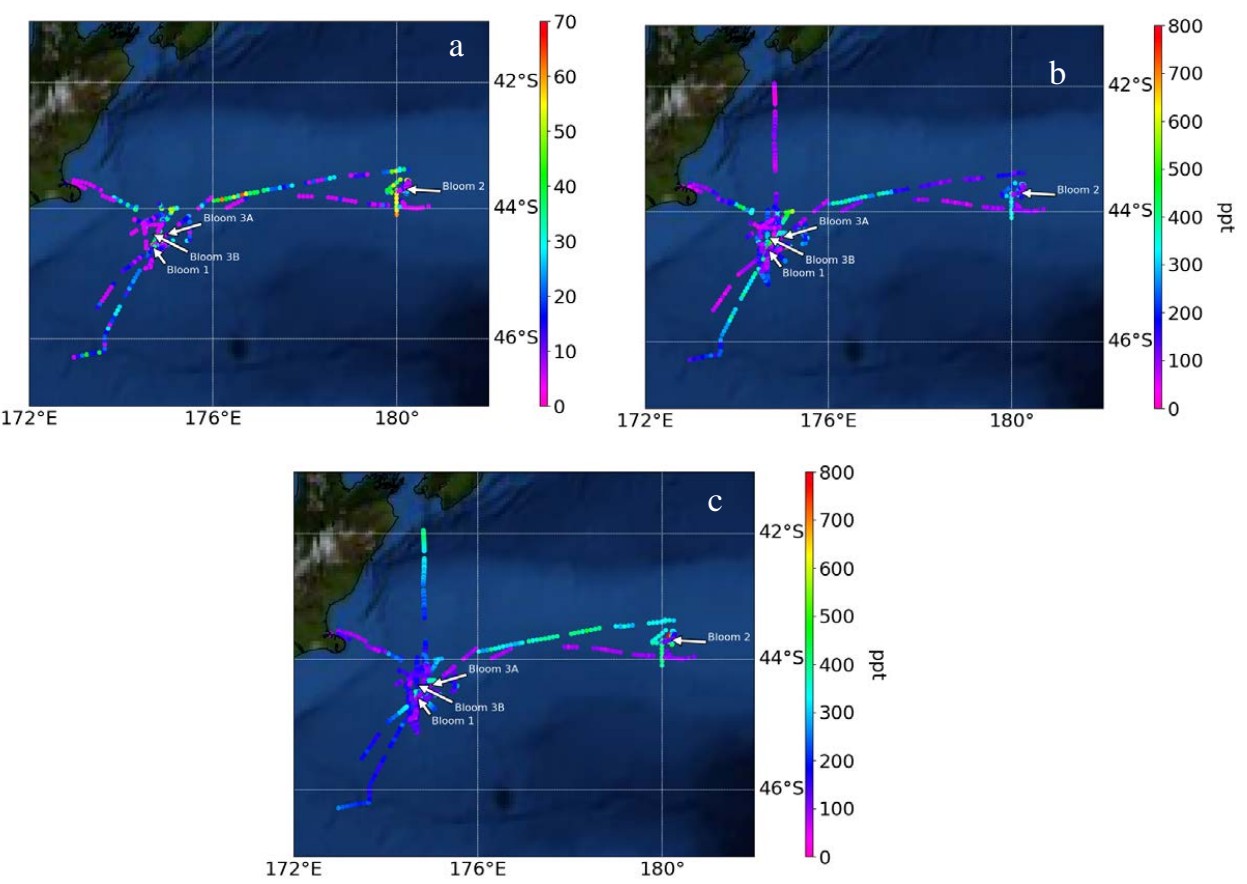

**Fig 3 Atmospheric mixing ratios of (a)MeSH$_a$, (b) DMS$_a$ and c) acetone$_a$ as function of the voyage track. Location of the blooms are shown.**

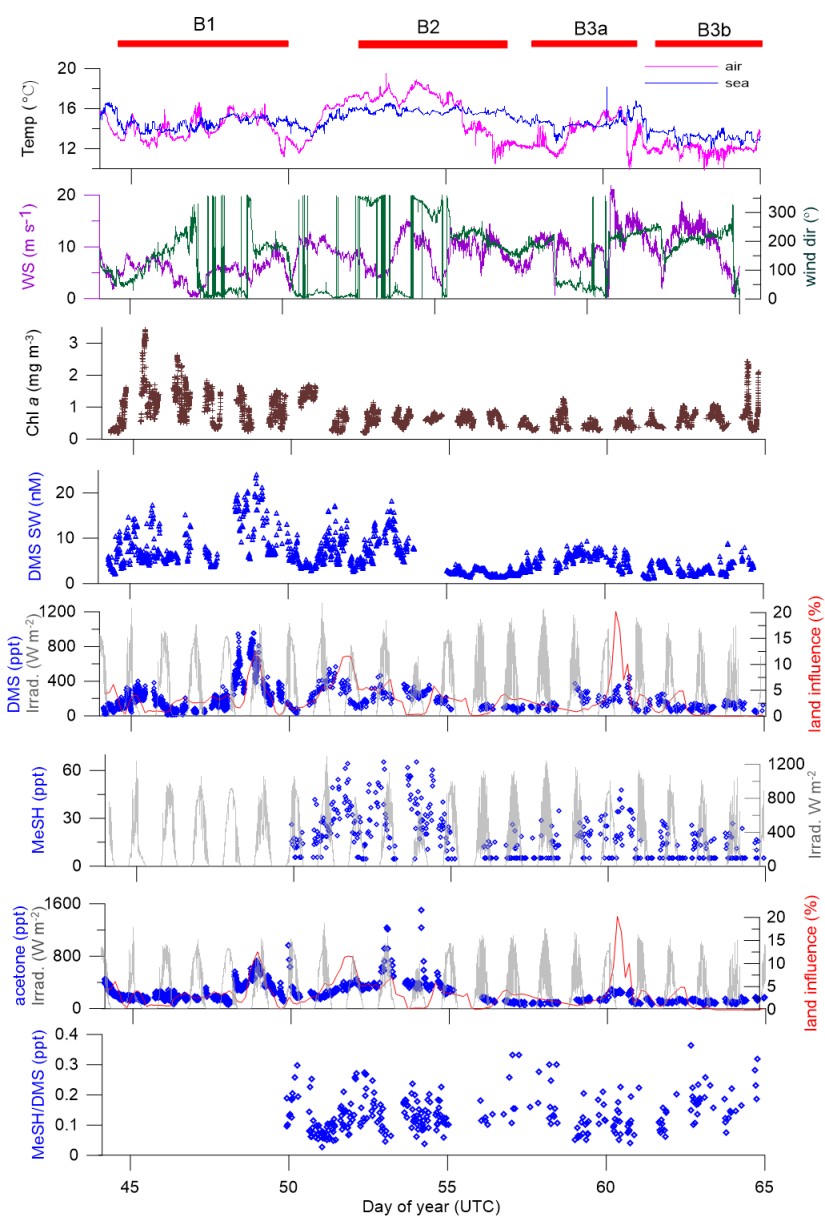

2 **Figure 4 -times series of measurements during the SOAP voyage according to DOY. Atmospheric DMS and MeSH**
3 **measurements below detection limit have had half detection limit substituted.**





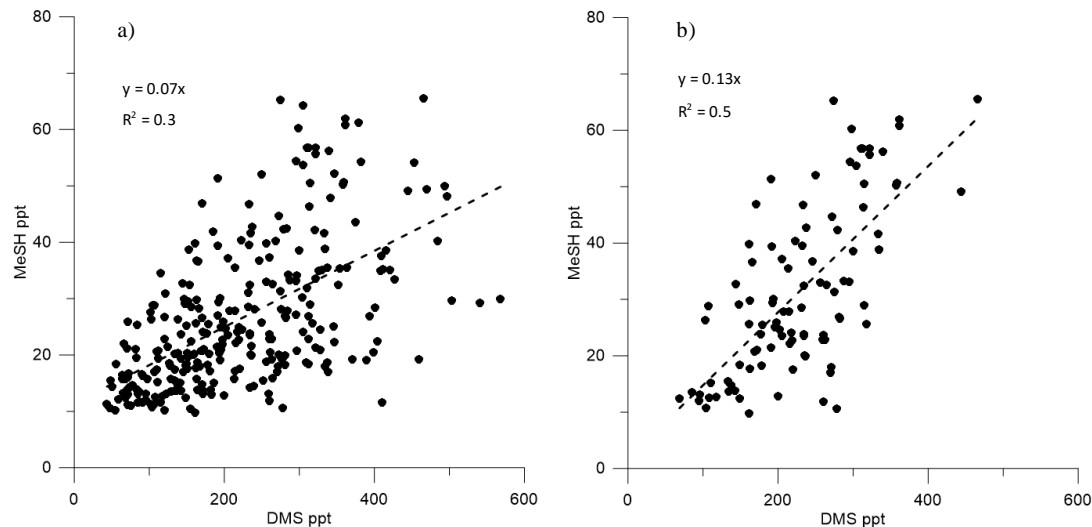

2      **Fig 5. Correlation between a) DMS$_a$ and MeSH$_a$ all data (DOY 49 onwards), b) DMS$_a$ and MeSH$_a$ bloom (B2) only**





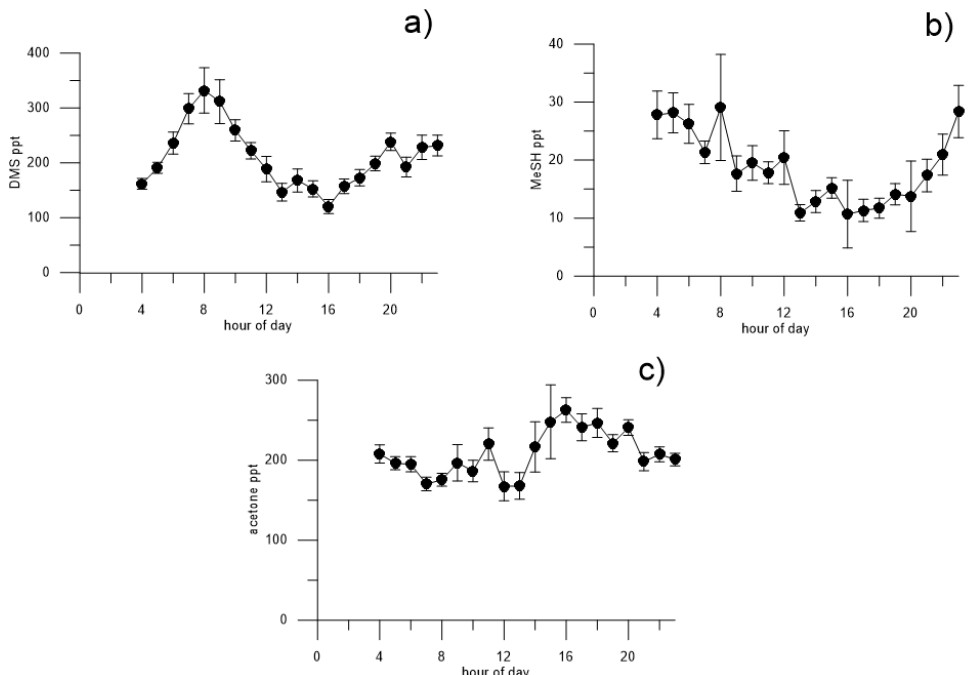

**Fig 6. Diurnal cycles of a) DMS$_a$, b) MeSH$_a$, c) acetone$_a$ with land influenced data removed. Average values from 0:00-
3:00 are excluded because of lower data collection during this period, due to calibrations and zero air measurements**

