# Peer review of "Methanethiol, dimethyl sulfide and acetone over biologically"

_Atmospheric Chemistry and Physics, 2019_

## Referee Comment (RC1) · Cathleen Schlundt (Referee) · 14 Nov 2019

General comments: The study by Sarah J. Lawson et al. "Methanethiol, dimethyl sulfide and acetone over biologically productive waters in the SW Pacific Ocean" is original, interesting, well written and an important contribution to the current knowledge of marine trace gases. The authors present an high resolution dataset in an undersampled remote marine region and explain the variation of the trace gases in the context of biogeochemical parameters. Furthermore, they present an intercomparsion of DMS measruements performed with three different independent analytical systems. I recommend to publish the paper with only a few minor corrections: Abstract Page 1, Line

21: You talked about "A positive flux of MeSH...". Can you write instead of positive or negative flux the direction of the flux? Into or out of the ocean.

Results and Discussion Page7 Line34: You mention "atmospheric stability". Can you explain shortly what do you mean with atmospheric stability in the text? Page 9 Line 2: You give the ratio between MeSH and DMS. Can you also present the average? Page 9 Line10: You say:"MeSH levels during B1 were substantially higher than..." Can give an actual number how high the level was. The reader will then be able to compare it with the literature value you present in this sentence.

Table 4: Can you present errors of the ppt values? In the figure caption please say: "nocturnal buildup method (NBL)" and say what EC stands for.

A general question: Why do you had different intake hights for you CIMS and your PTR-MS during the cruise? Isn't it esier to have the inlet of both instruments at the same location for comparison? Can you discuss this in the text?

———————————————————

---

## Referee Comment (RC2) · Anonymous Referee #2 · 12 Dec 2019

Review "Methanethiol, dimethyl sulfide and acetone over biologically productive waters in the SW Pacific Ocean" by Lawson et al.

The study by Lawson et al. reports atmospheric measurements of three trace gases relevant for atmospheric chemistry from a heavily undersampled region, and relates them to environmental parameters. A major finding is the varying but significant efflux of methanethiol to the atmosphere, indicating an oceanic source larger than previously thought. In addition, the paper contributes an intercalibration of DMS measurements, which is important for data comparability and helps to improve quality of DMS data compilation in larger datasets. Hence, the data reported is very valuable. The study

is well presented, timely and great care is given to details of how measurements were obtained and how the results were derived from the data. The study is very well suited for publication in Atmospheric Chemistry and Physics after some minor issues have been addressed:

General comments:

- The study remains rather descriptive. Did you have a clear hypothesis for the atmospheric concentration of these gases in the three regimes, e.g. before and during the blooms, and related to the bloom dominating species? If yes, you could state this more clearly in the introduction, and refer to it in the conclusion. Maybe you could emphasize the gap of knowledge that your study addresses more clearly at the end of the introduction.

- On page 11, l. 13 you state that an inherent assumption of the nocturnal accumulation method is a well mixed boundary layer, but in section 3.1 you state that part of the differences between measurement systems for DMS comes from the different intake heights. Isn't that a contradiction? Is it possible to take the information of concentration difference of the two inlets (i.e. concentration gradient) into account for the nocturnal accumulation method?

Specific comments:

Abstract:

p.1, l. 21: can you say "local time" after 16:00 hrs or say how many hours after local noon in order to make it more clear?

Introduction:

p. 2, l. 11: Could you provide a reference for the 17%?

Could you add a short sentence to the last paragraph about what was the aim of the study and what was the motivation to conduct this study in this area?
Methods:

p. 5, l. 23: Could you state whether you expect losses for acetone and MeSH, or why not (since they were not tested)? I am wondering if there are other studies showing the stability, which would support your results.

Results and discussion:

p. 8, l. 9ff you state that prior to day 47, the difference between the two measurement systems comes from calibration or undefined other differences. Can you further specify these differences? Or could it be that there are large uncertainties in the Smith equation, given that the trend in the differences between the two inlets decreased over time, but the absolute magnitude of the Smith-correction is not enough (but maybe carries uncertainties that cover the remaining differences?).

p. 10, l. 38ff: Could you discuss the physical control of the atmospheric concentration, e.g. would a breakdown of the boundary layer and an intrusion of free tropospheric air carrying less DMS/MeSH/acetone influence your measured concentration and your diurnal cycle as well?

p. 10, l 39ff: The finding of the differences in diel cycles between DMS and MeSH is interesting, since you state that both of them are removed by oxidation with OH. Do you attribute the remaining differences (e.g. increase in concentration during early morning for DMS but not for MeSH) to different production pathways, or to other additional sinks or physical processes that differ between those gases?

p. 12, section 3.4: Often you report potential explanations for your correlations, but you do not discuss them in detail (e.g. l. 22ff, l. 34ff). Could you be more specific here or derive further hypothesis/what needs to be tested specifically to prove whether this suggestion is likely/unlikely?

Implications and conclusion

My comments to this section is mainly reflected in the first general comment. Can

you use your data or the ratio of MeSH and DMS to derive a hypothesis under which environmental conditions the reaction pathway from DMSP favours DMS or MeSH formation? Or wouldn't that be reflected in the atmospheric data?

Figures

I think figure 6 would benefit from combining all diel cycles in one yy-axis plot (DMS and acetone on one axis and MeSH on the other) – so that one can compare the diel cycles together, as this is an aim of the study (p. 4, l. 6).

---

## Author Response (AR1)

**Author response to referees on "Methanethiol, dimethyl sulfide and acetone over biologically productive waters in the SW Pacific Ocean" by Sarah J. Lawson et al.**

We thank both referees for their careful reading of the manuscript and useful suggestions.
RC is referee comment and AR is author response.

**Minor corrections requested by Referee #1**

RC: Abstract Page 1, Line 21: You talked about "A positive flux of MeSH: : :". Can you write instead of positive or negative flux the direction of the flux? Into or out of the ocean.

AR: the text has been changed as suggested

RC: Results and Discussion Page 7 Line 34: You mention "atmospheric stability". Can you explain shortly what do you mean with atmospheric stability in the text?

AR: Atmospheric stability is a measure of the degree of vertical motion in the atmosphere, where $z/L= 0$ denotes neutral stability, $z/L >0$ denotes a stable atmosphere and $z/L < 0$ denotes an unstable atmosphere. This has been added to the text in this section.

RC: Page 9 Line 2: You give the ratio between MeSH and DMS. Can you also present the average?

AR: average is 0.14, this has been added to the text.

RC: Page 9 Line 10: You say:"MeSH levels during B1 were substantially higher than: : :" Can give an actual number how high the level was. The reader will then be able to compare it with the literature value you present in this sentence.

AR: text has been changed as follows to include levels observed, as shown below. Note, B1 was an error and has been changed to B2 in the revised text.

'MeSH$_a$ levels during B2 ranged from below detection limit (~10 ppt) up to 65 ppt (average 25 ppt), which is substantially higher than the only comparable measurements from the Drake Passage and the coastal and inshore waters west of the Antarctic Peninsula (3.6 ppt) (Berresheim, 1987).

RC: Table 4: Can you present errors of the ppt values?

AR: We have added error values to Table 4 as requested.

RC: In the figure caption please say:
"nocturnal buildup method (NBL)" and say what EC stands for.
AC: NBM and EC have been defined in Table 4 caption.

RC: A general question: Why do you had different intake hights for you CIMS and your PTR-MS during the cruise? Isn't it esier to have the inlet of both instruments at the same location for comparison? Can you discuss this in the text?

AR: the CIMS was deployed for DMS eddy covariance flux measurements and was housed in a container on the foredeck in order to sample from the bow mast. In contrast, the PTR-MS was deployed to scan the range of possible atmospheric VOCs. Due to space constraints, the PTR-MS was situated in the centre of the vessel (shelter deck), and sampled from the crows nest of the vessel.
We agree that it would have been desirable to co-locate the instruments and inlets, however the DMS comparison, while valuable, was not considered to be the primary aim of the voyage.

The following text has been added to the manuscript:

'The mesoCIMS was deployed primarily for DMS eddy covariance measurements, while the PTR-MS was deployed to measure atmospheric mixing ratios of a range of VOCs. As such, the mesoCIMS was situated on the foredeck and sampled from eddy covariance set up on the bow mast (12m a.s.l), while the PTR-MS was sited further back in the vessel and sampled from the crows nest (28m a.s.l.). Therefore, due to different intake heights, a further source of the difference between the PTR-MS and mesoCIMS measurements is likely due to vertical gradients in DMS caused by turbulent mixing......'

**Minor corrections requested by Referee #2**

General comments:
RC: The study remains rather descriptive. Did you have a clear hypothesis for the atmospheric concentration of these gases in the three regimes, e.g. before and during the blooms, and related to the bloom dominating species? If yes, you could state this more clearly in the introduction, and refer to it in the conclusion. Maybe you could emphasize the gap of knowledge that your study addresses more clearly at the end of the introduction.

AR: A main aim of this work was to explore the relationship between these gases and their chemical and biological ocean precursors, and to determine whether our findings were consistent with previous studies (where available, recognising that previous $MeSH_a$ measurements are extremely limited). The text has been changed as follows to highlight the relationships investigated and emphasise knowledge gaps:

' In this work, we present measurements of $DMS_a$, $MeSH_a$ and $acetone_a$, including the largest observed mixing ratios of $MeSH_a$ in the marine boundary layer to date. We explore the relationship between $DMS_a$, $MeSH_a$ and $acetone_a$ as well as the relationship with ocean biogeochemical parameters. In particular, we investigate links between $MeSH_a$ and its precursor DMSP for the first time. We explore whether variability in $acetone_a$ is linked to biogeochemistry, including warmer subtropical water and organic precursors such as CDOM as has been reported elsewhere. Given the large uncertainty in the oceanic budget of MeSH, we estimate the importance of MeSH as a source of atmospheric sulfur in this region and compare with other studies.'

We feel that the conclusion already summarises the relationships observed and how these findings fit in with other studies, however we have added the following text to the conclusion about the relationship between $MeSH_a$ and biogeochemistry:

'A correlation analysis of $MeSH_a$ and biogeochemical parameters was undertaken for the first time and showed that $MeSH_a$, as well as $DMS_a$ correlated with their ocean precursor, DMSP, and also correlated with seawater DMS ($DMS_{sw}$). The correlation of $MeSH_a$ with $DMS_{sw}$ likely due to a common ocean precursor of both gases (DMSP) which are produced via different pathways. '

RC: On page 11, l. 13 you state that an inherent assumption of the nocturnal accumulation method is a well mixed boundary layer, but in section 3.1 you state that part of the differences between measurement systems for DMS comes from the different intake heights. Isn't that a contradiction? Is it possible to take the information of concentration difference of the two inlets (i.e. concentration gradient) into account for the nocturnal accumulation method?

AR: The gradient at different intake heights is expected as a consequence of the surface flux of DMS into the atmosphere. There is always a logarithmic profile in concentration decreasing away from the surface for an emitted gas due to the vertical increase in eddy diffusivity away from the surface. This logarithmic layer is sometimes referred to as the surface layer to distinguish it from the whole boundary layer. The vertical concentration gradient is large very near the surface, but beyond 20 meters height or so, it is so small as to be undetectable by our methods. The time scale for mixing through that surface layer is very short (minutes or less)

so nocturnal emissions cannot accumulate there. Nocturnal emissions are fairly well mixed through the whole marine boundary layer (hundreds of meters) on a time scale of hours.

Specific comments:
RC: Abstract: p.1, l. 21: can you say "local time" after 16:00 hrs or say how many hours after local
noon in order to make it more clear?

AR: we have added local time

Introduction:
RC: p. 2, l. 11: Could you provide a reference for the 17%?

AR: have added reference to text: Lee and Brimblecombe, 2016

RC: Could you add a short sentence to the last paragraph about what was the aim of the
study and what was the motivation to conduct this study in this area?

AR: The following text has been added to the end of the introduction

'The Surface Ocean Aerosol Production (SOAP) voyage aimed to    investigate links between ocean biogeochemistry and aerosol and cloud processes in a biologically productive but under sampled region in the remote South West Pacific Ocean. '

Methods:
RC: p. 5, l. 23: Could you state whether you expect losses for acetone and MeSH, or why
not (since they were not tested)? I am wondering if there are other studies showing the
stability, which would support your results.

AR: for acetone, previous unpublished tests using ¼" PFA inlet tubing and ppb level mixtures of calibration gases
at 1-2 L/min with this PTR-MS showed acetone inlet losses of <5%. Similar results are expected in this study,
though a wider PFA tube diameter (3/8") and faster flow rate (10 L min$^{-1}$) was used.
MeSH has a similar molecular structure and physical properties to DMS at pH < 10 (section 3.2), so inlet surface
losses are likely to be similar to DMS.

The following text has been added
'Acetone inlet losses were tested previously using ppb level mixtures of calibration gases with PFA inlet tubing
and found to be <5%. MeSH has a similar structure and physical properties to DMS at pH < 10 (see Sect 3.2) and
so inlet losses are likely to be similar.
These small (<5%) losses this could lead to a minor underestimation in reported mixing ratios of DMS$_a$, acetone$_a$
and MeSH$_a$.'

Results and discussion:
RC: p. 8, l. 9ff you state that prior to day 47, the difference between the two measurement
systems comes from calibration or undefined other differences. Can you further specify
these differences? Or could it be that there are large uncertainties in the Smith equation,
given that the trend in the differences between the two inlets decreased over time,
but the absolute magnitude of the Smith-correction is not enough (but maybe carries
uncertainties that cover the remaining differences?).

AR: There was an error with the legend labelling on the last panel in Figure 1 which may have caused confusion – the label for observed and calculated difference were around the wrong way – this has been rectified in a new Figure 1. We apologise for this.

We believe that prior to DOY 47 the difference is most likely due to calibration or other instrument issues because the relative difference between instruments is relatively constant (~50%) until DOY 47, then it drops suddenly to an average of ~20%, albeit with some higher relative difference values on DOY 48. While instrument parameters and calibration responses were carefully examined during this period, we could not find a reason for the sudden increase in agreement between instruments on DOY 47. But similarly, there is no sudden change in the environmental parameters that would lead to a better agreement at the two inlet heights from DOY 47 onwards – for example the sudden improved agreement between DOY 47 – 50 would need to be driven by a weaker DMS source and a more well mixed atmosphere, whereas the DMS source (flux) increases over this period and the wind speed and stability mixing remains similar to the period prior to DOY 47.

The following text has been added:

'The reason for the improved agreement between mesoCIMS and PTR-MS at DOY 47 is unlikely due to a decrease in the DMS concentration gradient (Fig. 1 bottom panel), but is more likely due to changes in instrument calibration or other differences. However careful inspection of the instrument parameters, configurations and calibration responses prior to DOY 47 did not identify the cause of the disagreement.'

RC: p. 10, l. 38ff: Could you discuss the physical control of the atmospheric concentration, e.g. would a breakdown of the boundary layer and an intrusion of free tropospheric air carrying less DMS/MeSH/acetone influence your measured concentration and your diurnal cycle as well?

AR: Yes, entrainment of air from the free troposphere could lead to sampling of air with lower mixing ratios of short-lived DMS and MeSH. However for acetone, which is longer lived and has significant terrestrial sources, free tropospheric air could potentially be enhanced or depleted in acetone compared to MBL air, depending on the origin of the air.

The following text has been added:

'An additional factor which may influence the measured mixing ratios of $DMS_a$, $MeSH_a$ and $acetone_a$ is entrainment of air from the free troposphere into the MBL. For short-lived DMS and MeSH (Table 2), free tropospheric air is most likely to be depleted in these gases compared to air sampled close to the ocean surface. Acetone is relatively long lived (Table 2) and has significant terrestrial sources (Fischer et al., 2012), and so depending on the origin of the free tropospheric air, could be enhanced or depleted relative to MBL air.

RC: p. 10, l 39ff: The finding of the differences in diel cycles between DMS and MeSH is interesting, since you state that both of them are removed by oxidation with OH. Do you attribute the remaining differences (e.g. increase in concentration during early morning for DMS but not for MeSH) to different production pathways, or to other additional sinks or physical processes that differ between those gases?

AR: we agree that this warrants discussion. The following text has been added:

'The decoupling of the DMS and MeSH diurnal cycles between 4:00 – 8:00 hrs, with DMS increasing and MeSH decreasing, is likely due to the differing production pathways as well as possibility additional sinks for MeSH in the ocean during this time. This time period may also be influenced by mapping areas with lower $DMS_{sw}$ overnight and stationing the vessel over blooms with high $DMS_{sw}$ from 8:00 hrs each day, as described above.'

RC: p. 12, section 3.4: Often you report potential explanations for your correlations, but you do not discuss them in detail (e.g. l. 22ff, l. 34ff). Could you be more specific here or derive further hypothesis/what needs to be tested specifically to prove whether this suggestion is likely/unlikely?

AR: To be more specific about what needs to be tested to investigate the lack of correlation between DMSPd and MeSH (l. 22ff), the following text has been added

'whether destruction of MeSH via organic and particulate sinks is responsible for the lack of correlation with DMSPd could be further investigated in incubation experiments.'

AR: In terms of specific suggestions to investigate acetone$_a$ and biogeochemical correlations (eg . l. 34ff and others); In this work correlating acetone$_a$ with biogeochemical parameters is used to identify possible ocean sources of acetone. A limitation of this approach as described in the text is that acetone is long-lived and atmospheric variability may be related to sources other than the ocean. While acetone$_a$ data with terrestrial influence has been removed from this analysis, seawater acetone measurements would be needed to completely remove the influence of non- ocean sources and could be correlated with biogeochemical parameters.
Secondly, to be confident about the ocean source of acetone, for example whether photochemical, biological, or linked to cocclithophores, mesocosm or laboratory studies could be undertaken in which biogeochemical parameters and acetone production could be closely monitored.

The following text has been added to this section:

'Seawater acetone measurements would allow further elucidation of the relationships between acetone$_a$ and biogeochemical parameters identified in this study. More generally, mesocosm, or laboratory studies could be employed to identify the explicit sources and production mechanisms of these gases in Chatham Rise waters.'

Implications and conclusion
RC: My comments to this section is mainly reflected in the first general comment. Can you use your data or the ratio of MeSH and DMS to derive a hypothesis under which environmental conditions the reaction pathway from DMSP favours DMS or MeSH formation? Or wouldn't that be reflected in the atmospheric data?

AR: This atmospheric data cannot reliably be used to determine the ratio of MeSH/DMS in seawater. This is because of the different atmospheric lifetimes of DMS and MeSH which lead to different atmospheric ratios of MeSH/DMS depending on the age of the air mass/time since emission from the ocean. The best estimate of the MeSH/DMS seawater ratio is likely from the 3 nights when the flux was calculated by the nocturnal accumulation method (assuming no destruction by OH)-however data from these 3 nights is insufficient to correlate with environmental conditions.
A further consideration is that even if the atmospheric measurements could be used to determine the seawater MeSH/DMS ratio, the seawater ratio would not necessarily reflect the importance of different production pathways, because MeSH is lost much more quickly in seawater than DMS.

Figures
RC: I think figure 6 would benefit from combining all diel cycles in one yy-axis plot (DMS and acetone on one axis and MeSH on the other) – so that one can compare the diel cycles together, as this is an aim of the study (p. 4, l. 6).
AR: we plotted Fig 6 as suggested, however we think that it is difficult to see the behaviour of the individual gases as the plot is quite complex with 3 lots of series and associated error bars . As such we have kept the 3 diurnal plots separate.

[revised manuscript text omitted]

**Commented [LS(A2):** This is identical to previous plot, except that the legend on the bottom panel has been corrected.

**Figure 1 From top to bottom, wind speed and stability, DMS$_a$ measurements from mesoCIMS and PTR-MS, relative**
**difference (normalised to mesoCIMS) according to absolute wind direction, and absolute observed and calculated**
**difference between mesoCIMS and PTR-MS, taking into account the expected DMS concentration gradient (Eq. 1)**

[Figure]

**Fig 2 a) DMS$_a$ measured by mesoCIMS (x) and PTR-MS (y) b) mesoCIMS (x) and PTR-MS (y) DMS data corrected for the expected concentration gradient (observed PTR-MS DMS + calculated delta DMS)**

[Figure]

**Fig 3 Atmospheric mixing ratios of (a)MeSH$_a$, (b) DMS$_a$ and c) acetone$_a$ as function of the voyage track. Location of**
**the blooms are shown.**

[Figure]

**Figure 4 -times series of measurements during the SOAP voyage according to DOY. Atmospheric DMS and MeSH**
**measurements below detection limit have had half detection limit substituted.**

[Figure]

**Fig 5. Correlation between a) DMS$_a$ and MeSH$_a$ all data (DOY 49 onwards), b) DMS$_a$ and MeSH$_a$ bloom (B2) only**

[Figure]

Commented [LS(A3)]: The x axis label has been changed on this plot for clarity

**Fig 6. Diurnal cycles of a) DMS$_a$, b) MeSH$_a$, c) acetone$_a$ with land influenced data removed. Average values from 0:00-3:00 are excluded because of lower data collection during this period, due to calibrations and zero air measurements**

---

## Author Response (AR2)

Author comments to Editor:

Thank you for your comments. Please see responses below, EC is editor comment, AR is author response.

EC: Please write all units with negative exponents, e.g., m s-1

AR: units have been changed as requested (Table 4 and Table 5)

EC: P1, L39 – P2, L1-2 „VOCs may also impact air quality and human health, through their role in particle and ozone formation, and direct impacts through exposure." This sentence is grammatically incorrect (especially the last part does not seem to fit to the first), an anacoluthon. Please correct.

AR: the first two sentences have been replaced with:

"Volatile organic compounds (VOCs) are ubiquitous in the atmosphere and have a central role in processes affecting air quality and climate, via their role in formation of secondary organic aerosol and tropospheric ozone."

EC: P2, L7 I think a short explanation of the CLAW hypothesis would be useful to the reader.

AR: The following text has been added

"Since the publication of the CLAW hypothesis (Charlson et al., 1987), which proposed a climate feedback loop between ocean DMS concentrations and cloud droplet concentrations and albedo,….."

EC: P2, L12 "to be ~17% of the DMS source" It is strange to give the percentage without having given earlier the magnitude of the DMS source.

AR: The following text in italic has been added

"The ocean is a major source of reduced volatile sulfur gases  and the most well-studied of these is dimethyl sulfide (DMS) (CH3SCH3), *with a global ocean source of ~28 Tg S a$^{-1}$* (Lee and Brimblecombe, 2016)."

EC: P3, L21-22 "For oxygenated VOCs (OVOCs), whether the ocean acts as a source or a sink in a particular region depends on the concentration gradient between seawater and atmosphere (Carpenter et al., 2012)." This is self-evident. Please modify and add real info or delete.

AR: text deleted

EC: P6, L9 I am not sure if every reader knows what SLPM stands for. Please explain.

AR: changed to standard litres per minute

EC: P6, L38 pCO2 (2 as subscript)

AR: 2 subscripted

EC: P7, L2 variables instead of parameters? (a parameter is parameterized, right?)

AR: CTD measurements were described as parameters in the 2017 SOAP overview and preliminary results paper (Law et al., 2017, www. atmos-chem-phys.net/17/13645/2017/). Hence for consistency between manuscripts from the SOAP study we prefer to use the word parameter.

EC: P7, 7 §2.4 Many measured variables are listed. If these are used in the manuscript, the reader should know at least the precision, and if possible, the accuracy.

AR: A supplementary table has been provided to provide the measurement specifications and references. This supplementary table has been referred to in the text.

EC: P9, L16 I suggest to not use MDL as an abbreviation, as this only occurs three times, and the reader possibly has to search for the meaning.

AR: MDL replaced with minimum detectable limit

EC: P12, L36 Please define the subscripts of DMSP. This has not been done before.

AR: subscripts have been defined

EC: P12, L36 "The correlation of DMSa with DMSsw is clear …" Please be more specific.

AR: changed to …..'can be attributed to the positive flux of DMS out of the ocean….'

EC: P13, L10 Please define HMW

AR: Have defined this in Section 2.4 by adding the text (HMW) after High Molecular Weight
"In addition, organic parameters measured included High Molecular Weight (HMW) reducing sugars…"

EC: P13, L33-34 "from 30 to unity (Kettle 33 et al 2001), from 6-20 (Leck and Rodhe, 1991) and 2-5 (Kiene et al., 2017)." This info also occurs in the Intro. I don't think it is needed here.

AR: this text has been deleted

EC: P13, L34 Is "As such" necessary?

AR: Yes we think it is necessary because it links the previous sentences (describing the large amount of variability in MeSH flux and concentration ratios with DMS) with the statement that further studies are needed.

EC: P14 last paragraph: I am missing a judgement of which method produces the most reliable data. Differences of 20% are relatively high. I would expect some more ideas or indications about which is likely to deliver the best results. Even if the authors would not know for sure, it would be good to present the arguments for or against it. As it is now, the reader feels kind of lost after reading about these three methods with their particular results and hearing nothing about the reliability. Some few words about this should also be added to the Abstract.

AR: It should be noted that this study wasn't designed to determine which technique is more accurate and such a conclusion cannot be drawn from the data. However, the $R^2$ of the relationship between the gradient corrected PTR-MS data and mesoCIMS data was 0.69, supporting the idea that both instruments do a reasonable job of measuring variability in DMSa (given the different integrated versus discrete measurement approaches). The remaining observed differences of ~20% are likely due to differences in calibration scales used by the independently calibrated instruments.

The following text has been added to the manuscript:

Page 8 : the $R^2$ of the relationship between PTR-MS and mesoCIMS ambient measurements has been provided

The final paragraph of the conclusion has been rewritten to discuss the issue of different calibration scales and suggested further work:

"Finally, the SOAP voyage provided the opportunity to compare 3 independently calibrated DMS measurement techniques at sea (PTR-MS, mesoCIMS and GC-SCD). Agreement between the three techniques was generally good, however some systematic differences between the datasets were observed. Some of these differences were attributable to the near surface DMS gradient and the use of different inlet heights (28 and 12 m a.s.l for the PTR-MS and mesoCIMS respectively), as well as differing approaches of integrated versus discrete measurements. The remaining discrepancies were likely due to differences in calibration scales, suggesting that further investigation of the stability and/or absolute calibration of DMS standards used at sea is warranted."

The following text has been added to the abstract:

"Some differences were attributable to the DMSa gradient above the sea surface and differing approaches of integrated versus discrete measurements. Remaining discrepancies were likely due to calibration scales, suggesting that further investigation of the stability and/or absolute calibration of DMS standards used at sea is warranted."

AR: format changed

EC: P16, L28 35S-DMSP (format, superscript)

AR: subscripted

EC: P16, L54 Too many symbols in authors list

AC: corrected

EC: P17, L28 Please correct authors list

AC: corrected

EC: P18, L6 "H3O+(H2O)0.1" (please correct format)

AC: corrected

EC: P18, L12 Author: Jöckel (correct name)

AC: corrected

EC: P19, L4 write standard deviation, not std dev

AC: standard deviation written

EC: P19, L13 define CI here

AC: confidence interval written

EC: P20, L4 standard not std

AC: corrected

EC: P20, Table 6 What is TpCO2? I do not know of any variable with this symbol. Shouldn't this be just pCO2?

please use subscript for 2

AR: TpCO2 was inadvertently included and has been removed from this table.

EC: P20, Table 6 caption: Please add for which data and days these correlations apply.

AR: The caption of Table 6 now reads:

"Spearman rank correlations between acetone$_a$, DMS$_a$ and MeSH$_a$ and biogeochemical parameters, using data from the 14 February 2012 – 4 March 2012 (DMS$_a$ and acetone$_a$) and 20 February 2012 – 4 March 2012 (MeSH$_a$).

Correlations shown are significant at 95% confidence interval (CI). Correlation coefficient (and p-value) are shown. No entry indicates there was no correlation at 95% CI. Land influenced acetonea data excluded (see text for details)."

EC: P20, Table 6: Please add units to the variables and parameters. At no place in the manuscript units are given.

AR: units have been added to table 6

EC: P20, Table 6 Please be consequent with using capitals or not. Usually names of algae or nutrients do not get a capital.

AR: algae and nutrients changed to lower case

EC: P23 caption Figure 2: please add what the lines represent

AR: the following text has been added to Fig 2 caption:

"Dashed lines represent the reduced major axis regression, solid lines represent a 1:1 relationship.'

EC: P25, Figure 4 Please define abbreviations in the panels, like WS, Irrad, Chl

AR: these abbreviations have been defined

EC: P26, Figure 5 The notations $y=0.07x$ and $y=0.13x$ is not the full description of the lines. Please explain or complete.

AR: intercepts have been added

[revised manuscript text omitted]

Commented [LS(A4]: Same as figure above except UTC on x axis label

**Figure 1 From top to bottom, wind speed and stability, DMS$_a$ measurements from mesoCIMS and PTR-MS, relative difference (normalised to mesoCIMS) according to absolute wind direction, and absolute observed and calculated difference between mesoCIMS and PTR-MS, taking into account the expected DMS concentration gradient (Eq. 1)**

[Figure]

Fig 2 a) DMS$_a$ measured by mesoCIMS (x) and PTR-MS (y) b) mesoCIMS (x) and PTR-MS (y) DMS data corrected
for the expected concentration gradient (observed PTR-MS DMS + calculated delta DMS). Dashed lines represent the
reduced major axis regression and solid lines represent a 1:1 relationship.

[Figure]

**Fig 3 Atmospheric mixing ratios of (a)MeSH$_a$, (b) DMS$_a$ and c) acetone$_a$ as function of the voyage track. Location of**
**the blooms are shown.**

[Figure]

Figure 4 -times series of measurements during the SOAP voyage according to DOY. Atmospheric DMS and MeSH measurements below detection limit have had half detection limit substituted. WS = wind speed, wind dir = wind direction, Irrad. = irradiance, Chl a =chlorophyll *a*

[Figure]

**Fig 5. Correlation between a) DMS$_a$ and MeSH$_a$ all data (DOY 49 onwards), b) DMS$_a$ and MeSH$_a$ bloom (B2) only**

[Figure]

Fig 6. Diurnal cycles of a) DMSa, b) MeSHa, c) acetonea with land influenced data removed. Average values from 0:00-3:00 are excluded because of lower data collection during this period, due to calibrations and zero air measurements